# Permissive and instructive *Hox* codes govern limb positioning

Yajun Wang[1]*[†], Maik Hintze[1], Jinbao Wang[1,2], Hengxun Tao[1], Patrick Petzsch[3], Karl Köhrer[3], Longfei Cheng[1], Peng Zhou[1,4], Jianlin Wang[5], Zhaofu Liao[6], Xu-Feng Qi[6], Dongqing Cai[6], Thomas Bartolomaeus[7], Karl Schilling[8], Joerg Wilting[9], Stefanie Kuerten[1,10], Georgy Koentges[11], Ketan Patel[12], Qin Pu[1], Ruijin Huang[1,10]*

[1]Institute of Neuroanatomy, University of Bonn, Medical Faculty, Bonn, Germany; [2]School of Basic Medical Sciences, Ningxia Medical University, Yinchuan, China; [3]Biological and Medical Research Centre (BMFZ), Medical Faculty, Heinrich-Heine-University Duesseldorf, Düsseldorf, Germany; [4]Institute of Zoology, School of Life Sciences, Lanzhou University, Lanzhou, China; [5]College of Pastoral Agriculture Science and Technology, Lanzhou University, Lanzhou, China; [6]Key Laboratory of Regenerative Medicine, Ministry of Education, Department of Developmental and Regenerative Biology, Jinan University, Guangzhou, China; [7]Institute of Evolutionary Biology and Animal Ecology, Rheinische Friedrich-Wilhelms-Universität, Bonn, Germany; [8]Institute of Anatomy, Department of Cell Biology, University of Bonn, Medical Faculty, Bonn, Germany; [9]Deparment of Anatomy and Cell Biology, University Medical School Goettingen, Goettingen, Germany; [10]University Hospital Bonn, Bonn, Germany; [11]Laboratory of Systems Biomedicine and Evolution, School of Life Sciences, University of Warwick, Coventry, United Kingdom; [12]School of Biological Sciences, University of Reading, Reading, United Kingdom

*For correspondence:
yajun0809@163.com (YW);
ruijin.huang@uni-bonn.de (RH)

Present address: [†]Shandong Medical And Pharmaceutical University, Yantai, China

Competing interest: The authors declare that no competing interests exist.

## eLife Assessment

This **important** study provides the first putative evidence that alteration of the Hox code in neck lateral plate mesoderm is sufficient to induce ectopic development of forelimb buds at neck level. The authors use both gain-of-function (GOF) and loss-of-function (LOF) approaches in chick embryos to test the roles of Hox paralogy group (PG) 4-7 genes in limb development. The GOF data provide strong evidence that overexpression of Hox PG6/7 genes are sufficient to induce forelimb buds at neck level. However, the experiments using dominant negative constructs are lacking some key controls that are needed to demonstrate the specificity of the LOF effect rendering the work as a whole **incomplete**.

**Abstract** The positioning of limbs along the anterior-posterior axis varies widely across vertebrates. The mechanisms controlling this feature remain to be fully understood. For over 30 years, it has been speculated that *Hox* genes play a key role in this process, but evidence supporting this hypothesis has been largely indirect. In this study, we employed loss- and gain-of-function *Hox* gene variants in chick embryos to address this issue. Using this approach, we found that *Hox4/5* genes are necessary but insufficient for forelimb formation. Within the *Hox4/5* expression domain, *Hox6/7* genes are sufficient for reprogramming of neck lateral plate mesoderm to form an ectopic limb bud, thereby inducing forelimb formation anterior to the normal limb field. Our findings demonstrate that the forelimb programme depends on the combinatorial actions of these *Hox* genes. We propose that during the evolutionary emergence of the neck, *Hox4/5* provides permissive cues for forelimb

formation throughout the neck region, while the final position of the forelimb is determined by the instructive cues of *Hox6/7* in the lateral plate mesoderm.

## Introduction

The spatial development of vertebrate tissues is regulated by *Homeobox* (*Hox*) genes (*Duboule, 2022*; *Iimura and Pourquié, 2006*; *Zakany and Duboule, 2007*). A huge literature evidences that *Hox* genes determine the development and patterning of the vertebrate axial skeleton (reviewed in *Burke, 2000*). Mutations in *Hox* genes can lead to homeotic transformations, where one type of vertebra is transformed into another (*Böhmer, 2017*). Vertebrate limbs emerge at specific axial levels along the anterior-posterior (AP) axis, with precise positioning varying significantly across species (*Burke et al., 1995*). These characteristics make limb positioning a valuable experimental model for studying the mechanisms regulating positional information (*Zakany and Duboule, 2007*). Despite variable numbers of cervical vertebrae between species, the pectoral fin or forelimb is always located at the cervical-thoracic boundary. The mechanisms underpinning the positioning of vertebrate forelimbs remain to be fully elucidated.

While *Hox* gene misexpression causes substantial alterations in vertebrae identity (*Garcia-Gasca and Spyropoulos, 2000*; *Horan et al., 1995*; *Jeannotte et al., 1993*; *Ramfrez-Solis et al., 1993*), only minor changes in limb development have been observed in *Hox* gene mutants (*Rancourt et al., 1995*). It is therefore unclear whether, for example, the abnormal limb that develops in *Hoxb5* mutants represents a true shift in the limb field or rather a shoulder girdle defect causing the forelimb to appear 'shrugged' anteriorly. In fact, whereas knockdown of the complete paralogous group Hox5 genes results in changes in limb patterning, it does not result in a positional shift of the forelimb (*Xu et al., 2013*).

Moreover, interpretation of the effects of *Hox* genes on limb positioning in global knockouts is fraught by the fact that this not only affects lateral plate mesoderm patterning, but invariably also vertebrae-forming mesoderm and vertebrate identity. Yet normal vertebrate identity is required as a reference for defining limb positions. Ideally, limb positioning should be investigated by limiting the manipulation of *Hox* expression to the limb-forming mesoderm, without altering vertebral positional identity.

The initiation of the forelimb programme is marked by *Tbx5* expression in the LPM, which is functionally required for pectoral fin formation in zebrafish and forelimb formation in chicken and mice (*Hasson et al., 2007*; *Rallis et al., 2003*; *Takeuchi et al., 2003*). However, the forelimb-forming potential is present in mesodermal cells at the cervico-thoracic transitional zone long before the activation of *Tbx5* expression (*Chaube, 1959*; *Moreau et al., 2019*). This has led to the notion that cells first acquire positional identity through the expression of *Hox* genes, followed by a developmental programme guided by their positional history (*Duboule, 2022*; *Iimura and Pourquié, 2006*; *Zakany and Duboule, 2007*).

The positional identity of future limb-forming cells of the LPM is coded by the nested and combinatorial expression of *Hox* genes (*Duboule and Dollé, 1989*; *Kessel and Gruss, 1991*). Only a few studies have investigated how this *Hox* code translates to *Tbx5* expression in the prospective forelimb region and thus regulates forelimb positioning (*Moreau et al., 2019*). During gastrulation, the collinear activation of *Hox* genes begins in the epiblast, conferring AP identity to the paraxial mesoderm (*Duboule, 2022*; *Iimura and Pourquié, 2006*). A similar mechanism regulates the anteroposterior patterning of the LPM, which gives rise to limbs. For instance, *Hoxb4*-expressing cells emigrating from the posterior part of the primitive streak form the LPM in the neck. Subsequently, *Hoxb4* activates *Tbx5* expression within this LPM domain. The limb positioning is thus regulated by *Hox* genes in two phases (*Minguillon et al., 2012*; *Moreau et al., 2019*; *Nishimoto et al., 2014*). During the first phase, *Hox*-regulated gastrulation movements establish the forelimb, interlimb, and hindlimb domains in the LPM. In the second phase, a *Hox* code regulates *Tbx5* activation in the forelimb-forming LPM (*Minguillon et al., 2012*; *Moreau et al., 2019*; *Nishimoto et al., 2014*). The forelimb-forming *Hox* code is considered to be constituted by both repressing and enhancing *Hox* genes: caudal *Hox* genes, including *Hox9*, suppress and thus limit *Tbx5* expression, whereas rostrally expressed *Hox* genes activate *Tbx5* expression (*Minguillon et al., 2012*; *Nishimoto et al., 2014*). To date, *HoxPG4* and *PG5* genes are considered as activators of *Tbx5* (*Minguillon et al., 2012*; *Nishimoto et al., 2014*). The

function of *PG6* and *PG7* genes (*Becker et al., 1996a*; *Becker et al., 1996b*), which are also prominently expressed in the forelimb region, has so far not been analysed.

Here, we aimed to investigate which *Hox* genes act to position the anterior limb in chicks. We present evidence that wing position is controlled by a permissive signal governed by *HoxPG4/5* which demarcates a territory where it can form. However, an additional instructive cue mediated by *HoxPG6/7* genes within the permissive region is required for forelimb formation. Our study is the first to show that neck LPM can be re-specified to form limb.

## Results

### HoxPG4–7 are required for the forelimb formation

The expression domain of *HoxPG6/7*, like that of *HoxPG4/5*, overlaps with the forelimb field, suggesting they might activate *Tbx5* expression. To untangle the roles of individual members of the *HoxPG4/5/6/7*, we performed loss-of-function experiments in chick embryos. We focused on the A-cluster of *HoxPG4/5/6/7*, using specifically generated dominant-negative (DN) forms to suppress the signalling function of each target *Hox* gene. The DN variants lack the C-terminal portion of the homeodomain, rendering them incapable of binding to the target DNA while preserving their function of binding transcriptional specific co-factors (*Denans et al., 2015*; *Gehring et al., 1990*). The specificity and effectiveness of this DN strategy have been further validated in a recent study showing that expression of a Hoxb4 DN construct led to a reduction in the Tbx5 expression domain during limb induction (*Moreau et al., 2019*), consistent with a specific loss of Hoxb4 function. Plasmids expressing DN *Hoxa4*, *a5*, *a6*, or *a7* were electroporated into the dorsal layer of LPM in the prospective wing field, from which the wing mesoderm originates in Hamburger-Hamilton stage (HH) 12 chick embryos (*Hamburger and Hamilton, 1951*; *Figure 1a and b*). After 8–10 hr, embryos reached HH14 when expression from the transfected DN constructs was detectable in the wing field of the transfected (right) side signified by enhanced green fluorescent protein (EGFP) expression also encoded by these plasmids (*Figure 1c*).

*Tbx5*, as the first gene indicating activation of the forelimb-forming programme, starts to be expressed in the forelimb field of normal chick embryos at HH13 (http://geisha.arizona.edu). Therefore, we analysed *Tbx5* expression following inhibition of *Hoxa4/5/6/7* at HH14. Expression of *Tbx5* in the wing field transfected with DN plasmids for any of these genes was consistently lower than in the contralateral (control) side (*Figure 1B–E*, *Table 1*). The down-regulation of *Tbx5* expression by all four DN forms of *Hoxa4/5/6/7* shows a previously unknown requirement of *PG6* and *PG7 Hox* genes for the activation of *Tbx5* during forelimb induction and confirms the previously reported *Tbx5* activating effects of *PG4* and *PG5 Hox* genes (*Nishimoto et al., 2014*).

*Tbx5* is required for the activation of *Fgf10* in the mesoderm (*Cohn et al., 1995*; *Min et al., 1998*; *Sekine et al., 1999*; *Young et al., 2019*). *Fgf10* subsequently induces *Fgf8* expression in the overlying ectoderm to initiate forelimb outgrowth (*Barrow et al., 2003*). The two genes form a positive feedback loop to ensure formation of the apical ectodermal ridge (AER), which ultimately regulates sustainable outgrowth and patterning (*Crossley et al., 1996*; *Min et al., 1998*). We analysed their expression at HH18–19. DN inhibition of any of the *Hoxa4/5/6/7* genes reduced the expression levels and domains of *Fgf10* (*Figure 1G–J*; *Table 1*) and *Fgf8* (*Figure 1L–O*; *Table 1*). The consequence of these manipulations on the outgrowth of the wing bud was analysed at HH22, when the wing bud develops a nearly square shape. After inhibition of HOX proteins, the form of the target wing buds was altered and their size was decreased. In some cases, the anteroposterior extent of the wing bud was also remarkably reduced (*Figure 1Q–T*). To quantify the effect of Hox inhibition on wing bud development, we measured the proximal-distal (P-D) elevation of the electroporated wing bud above the trunk lateral surface, compared to the contralateral non-electroporated control wing bud (*Figure 1U*).

Electroporation of plasmid-free solution and an EGFP-encoding plasmid caused only minimal reduction compared to their contralateral wing bud, indicating low developmental toxicity of the procedure of electroporation itself (*Figure 1U*).

Interference with the action of the representative A-cluster *Hox* genes indicates that *Hox* genes from all four paralogous groups (*PG4*, *PG5*, *PG6*, and *PG7*) impinge on the forelimb programme and

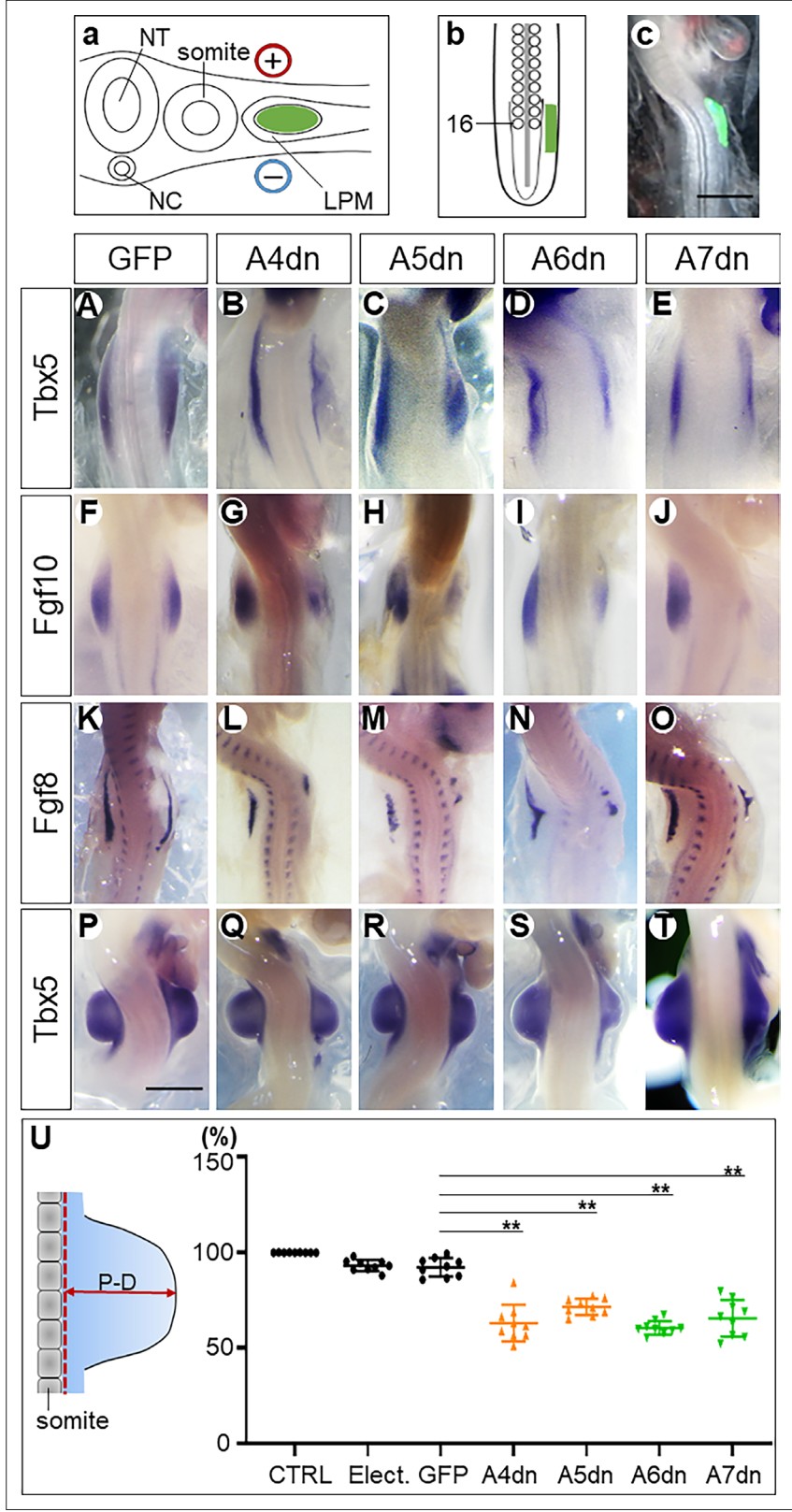

**Figure 1.** Hoxa4/a5/a6/a7 genes are necessary for wing bud formation. Schemes showing the electroporation in transverse section (**a**) and in the dorsal view (**b**). The somite 16 is marked (**b**). Successful transfection of plasmids as verified by EGFP expression (**c**). The *dn Hox* genes downregulated the expression of Tbx5 (**B–E**), Fgf10 (**G–J**), and Fgf8 (**L–O**) and inhibited wing bud formation at the ipsilateral (right) side (**Q–T**). **A–E**: HH14; **F–O**: HH18–19; **P–**

*Figure 1 continued on next page*

*Figure 1 continued*

**T**: HH22; scale bars in **c** (for **c**, **A–O**) and in **P** (for **P–T**): 500 µm. The proximodistal (**P–D**) distance (left in **U**) of wing buds is significantly reduced in *Hox dn*-expressing wing buds compared to EGFP electroporated wing buds (right in **U**). The scheme on the left-hand side shows how measurements were made. Red dotted line: baseline of the wing bud; CTRL: normal control wing buds without any operation; Elect.: wing buds after electroporation without constructs; GFP: wing buds after electroporation with EGFP-expressing constructs; A4dn, A5dn, A6dn, A7dn: wing buds after electroporation with *dn* Hoxa4/5/6/7 expressing constructs, respectively. Each dot represents one embryo; error bars represent mean ± SEM. \*\*p<0.01.

should be considered part of the activating *Hox* code for forelimb development. Overall, the effect of each *Hox* gene is limited, suggesting they act in a combinatorial, and possibly redundant fashion.

## *Hox6/7*, but not *Hox4/5*, are sufficient to reprogram neck to wing mesoderm

We next investigated the role of *HoxPG6/7* during forelimb fate determination. We hypothesised that if *HoxPG6/7* are (an) integral and necessary part(s) of the forelimb *Hox* code, their ectopic expression in a non-limb region, similar to the limb-inducing activity of FGFs (*Cohn et al., 1995*), should induce forelimb formation. In the present study, the neck was chosen as the non-limb region.

When A-cluster genes were electroporated at HH11–12 into the dorsal LPM at the level of somites 10–14 (anterior to the wing field) (*Figure 2a*), strong expression could be verified anterior to the cognate wing field by in situ hybridisation (ISH) 12 hr after electroporation (*Figure 2b–e*), indicating successful expression of *Hox* gene constructs.

The anterior expression domain of *HoxPG6/7* overlaps with the forelimb field but does not extend into the neck region. Electroporation of constructs expressing *Hoxa6/7* into the neck mesoderm caused their ectopic expression anterior to the forelimb field (*Figure 2d and e*). This induced ectopic expression of *Tbx5* in this region anterior to the cognate wing field (*Figure 2D and E*). By 48 hr re-incubation, a bulge appeared in the neck region transfected with *Hoxa6/7*. This bulge expressed the forelimb master gene *Tbx5*, and expression strength was similar to that of the natural wing bud (*Figure 2I and J*). Hence, it can be considered as an ectopic wing bud in the neck.

In contrast to ectopic expression of *Hoxa6/7* in the neck region, overexpression of *Hoxa4/5* (*Figure 2b and c*) by electroporating this region with *Hoxa4/5* coding plasmids did not extend *Tbx5* expression anteriorly (*Figure 2B and C*), indicating that no wing-forming mesoderm was ectopically induced in the neck by *Hoxa4/5* overexpression. Consequently, no structure emerged from the neck anterior to the endogenous wing bud after 48 hr of re-incubation (*Figure 2G and H*). These results demonstrate that *Hoxa4* and *Hoxa5* are insufficient, whereas *Hoxa6* and *Hoxa7* are sufficient to specify wing mesoderm in the neck region.

To ascertain whether other members of *HoxPG6/7* share the forelimb-inducing activity of the A-cluster genes, plasmids encoding full-length *Hoxb6* and *Hoxc6*, as well as *Hoxa7* and *Hoxb7*, were ectopically expressed in the region anterior to the wing field. After 48 hr, we observed either an anteriorly extended wing bud or a separated bud in the neck anterior to the endogenous wing bud (n=226/440, *Table 2*). The efficiency of transfection and transcription was monitored by assessing

**Table 1.** Dominant-negative expression of Hoxa4/a5/a6/a7 down-regulated gene expression.
The numbers of embryos with an unambiguous effect and the total number of embryos analysed are given (effect/total number analysed).

| | Tbx5 | | | Fgf10 | | | Fgf8 | | |
|---|---|---|---|---|---|---|---|---|---|
| | Expression level decreased | Expression domain shortened | Both down-regulation | Expression level decreased | Expression domain shortened | Both down-regulation | Expression level decreased | Expression domain shortened | Both down-regulation |
| A4dn | 15/17 | 12/17 | 10/17 | 10/11 | 9/10 | 9/10 | 10/10 | 6/10 | 6/10 |
| A5dn | 6/6 | 5/6 | 5/6 | 5/6 | 5/6 | 5/6 | 4/6 | 4/6 | 4/6 |
| A6dn | 6/6 | 6/6 | 6/6 | 6/6 | 5/6 | 5/6 | 5/5 | 5/5 | 5/5 |
| A7dn | 5/8 | 4/8 | 5/8 | 6/8 | 6/8 | 6/8 | 6/6 | 5/6 | 5/6 |

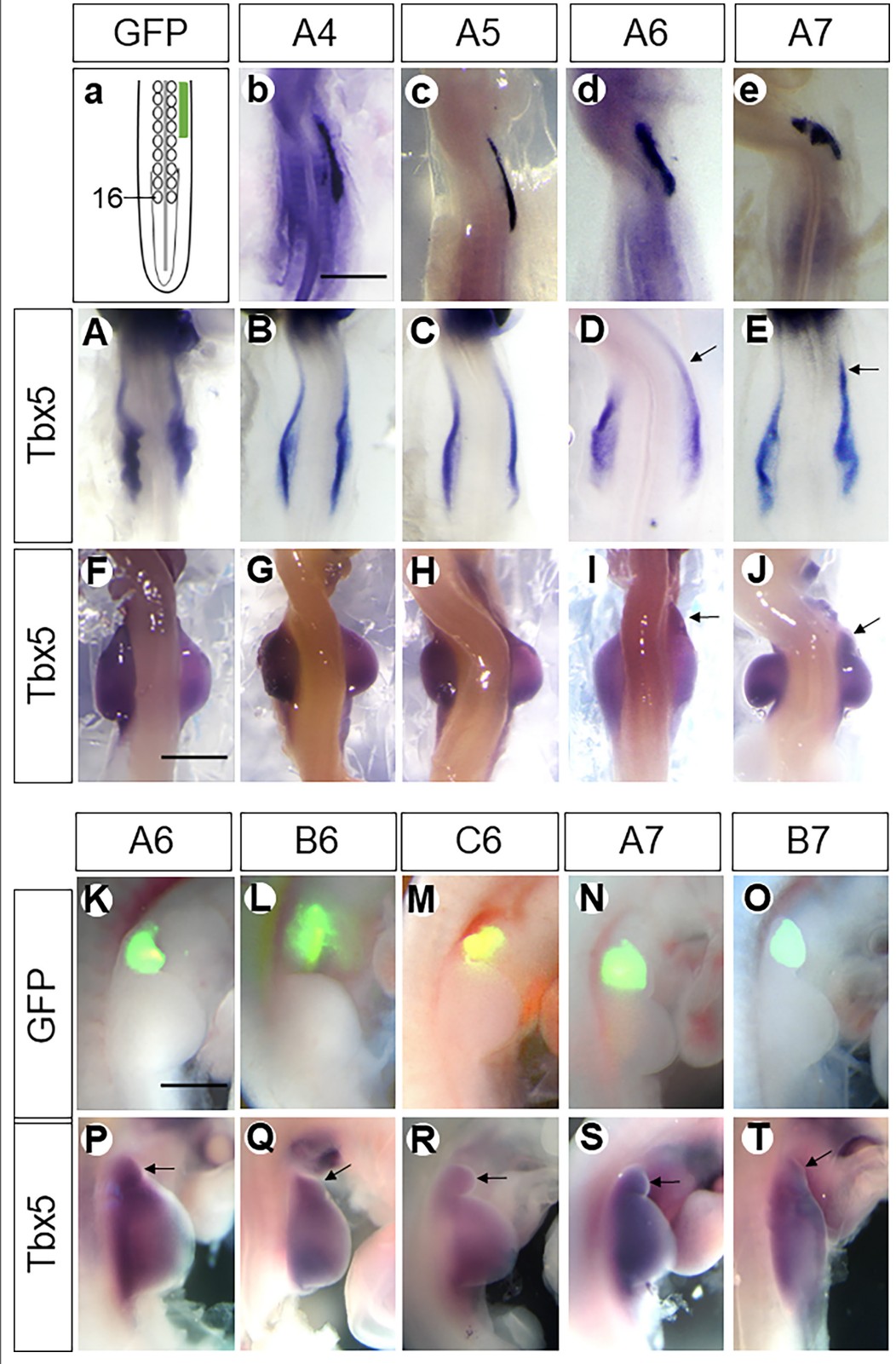

**Figure 2.** *Hoxa6/a7,* but not *Hoxa4/a5,* are sufficient to induce a neck wing bud. Scheme showing electroporation of the neck region in the dorsal view (**a**). The somite 16 is marked. The expression domain of electroporated constructs is marked by a green bar. Expression of *Hoxa4* (**b**), *Hoxa5* (**c**), *Hoxa6* (**d**), *Hoxa7* (**e**) in the LPM anterior to the wing field after electroporation with the respective plasmids as documented by in situ hybridisation. Whereas

*Figure 2 continued on next page*

*Figure 2 continued*

ectopic cervical expression of *Hoxa6/a7* induced the anterior expression (indicated by arrows) of *Tbx5* (**D, E, I, J**), overexpression of *Hoxa4/a5* did not induce anterior expression of it (**B, C, G, H**). Also, only Hoxa6 and Hoxa7, but not a4 or a5, resulted in the anterior extension of the wing bud (arrows in **I–J**). The ectopic wing buds (fused with or separated from the endogenous one) induced by *HoxPG6–7* are indicated by GFP fluorescence (**K–O**) and in situ hybridisation for *Tbx5* (arrows in **P–T**). **b–e** and **A–E**: HH14; **F–T**: HH22; scale bars in **b** (for **b–e** and **A–E**), in **F** (for **F–J**) and in **K** (for **K–T**): 500 μm. Arrows indicate induced wing buds (**P–T**).

EGFP expression from the plasmids used (*Figure 2K–O*), and their wing-inducing effect by screening induced *Tbx5* expression (*Figure 2P–T*). In more than half of the embryos, a separate wing bud, indicated by *Tbx5* expression, formed anteriorly to the endogenous wing bud (n=128/226, *Table 2*). In the remaining embryos, the endogenous wing bud appeared extended anteriorly (n=98/226, *Table 2*). These findings demonstrate that the ectopic formation of a wing bud in the neck is a consequence of the expression of all members of the HoxPG6/7 gene family.

Curiously, the induced wing buds did not grow distally to any great degree and remained small after 48 hr of re-incubation. To elucidate this phenomenon, RNA sequencing was used to compare gene expression in the induced wing buds with that of normal wing buds. Each group (*Figure 3A*) was comprised of four replicates. Ectopic expression of *Hoxa6* resulted in the up-regulation of multiple genes shared with normal wing buds, and the gene expression pattern in *A6*-induced wing buds was more similar to that of cognate wing buds than to that of native neck tissue (*Figure 3B and B'*). Gene Ontology (GO) biological process terms for 221 genes showed that the *A6*-induced bud closely resembles a normal wing bud (*Figure 3C*, *Table 3*). Functional categorisation revealed that 221 genes classified by GO biological process terms 'anterior/posterior pattern specification' ($p_{genuine}$ = $3 \cdot 2^{-10}$; $p_{induced}$ = $2 \cdot 5^{-10}$), 'proximal/distal pattern formation' ($p_{genuine}$ = $3 \cdot 7^{-9}$; $p_{induced}$ = $8 \cdot 7^{-9}$), 'regulation of transcription from RNA polymerase II promoter' ($p_{genuine}$ = $8 \cdot 4^{-11}$; $p_{induced}$ = $3 \cdot 2^{-8}$), 'embryonic skeletal system morphogenesis' ($p_{genuine}$ = $2 \cdot 0^{-9}$; $p_{induced}$ = $1 \cdot 3^{-5}$), and 'embryonic limb morphogenesis' ($p_{genuine}$ = $8 \cdot 1^{-9}$; $p_{induced}$ = $1 \cdot 4^{-4}$) were enriched in tissue of the genuine limb bud and in limb buds induced by *Hoxa6* overexpression (*Table 4*). In contrast, genes associated with the biological process terms 'cell adhesion' ($p$=$4 \cdot 3^{-19}$), 'extracellular matrix organisation' ($p$=$4 \cdot 2^{-15}$), 'transmembrane receptor protein tyrosine kinase signalling pathway' ($p$=$4 \cdot 8^{-13}$), 'positive regulation of kinase activity' ($p$=$1 \cdot 6^{-10}$), and 'multicellular organism development' ($p$=$3 \cdot 1^{-9}$) were overrepresented among the genes enriched in neck tissue (*Table 4*). These findings demonstrate that *Hoxa6* is sufficient for wing bud induction.

Although the wing programme in *A6*-bud revealed by *Tbx5* was initiated, the AER was not established. Expression of *Fgf10* was activated in the neck, resulting in the initiation of mesodermal outgrowth. However, its expression level was lower than that of the physiological wing-forming mesoderm (*Figure 3D–F*). In contrast, *Fgf8* was not induced in the ectoderm (*Figure 3D, G, and H*). Thus, the feedback loop between *Fgf10* and *Fgf8* was missing in the induced wing bud, and it failed to form an AER. Failure of the formation of functional AER is also indicated by the low levels of *Shh* expression in the induced wing bud as compared to the physiological wing anlage (*Fernandez-Guerrero et al., 2022*; *Lin and Zhang, 2020*). Without AER, the induced wing bud did not grow further. Further, the zone of polarising activity (ZPA) identified by the expression of *Shh* was not established (*Figure 3D and I*). Finally, we noted that the induced wing bud was dorsalised, as indicated by the strongly upregulated expression of Lmx1 (*Figure 3D and J*).

Taken together, we conclude that *HoxPG6/7* genes are sufficient for forelimb specification in the neck region. However, the induced wing bud is incapable of establishing the positive feedback loop

**Table 2.** HoxPG6/7 up-regulated wing bud formation in the neck region.
Number indicates the numbers of embryos in which a cervical extension of the wing bud, or a cervical wing bud separated from the normal wing bud could be observed.

| | A6 | B6 | C6 | A7 | B7 | Total |
|---|---|---|---|---|---|---|
| Extension | 31 | 7 | 5 | 37 | 18 | 98 |
| Separated | 45 | 10 | 10 | 51 | 12 | 128 |
| Total | 76 | 17 | 15 | 88 | 30 | 226 |

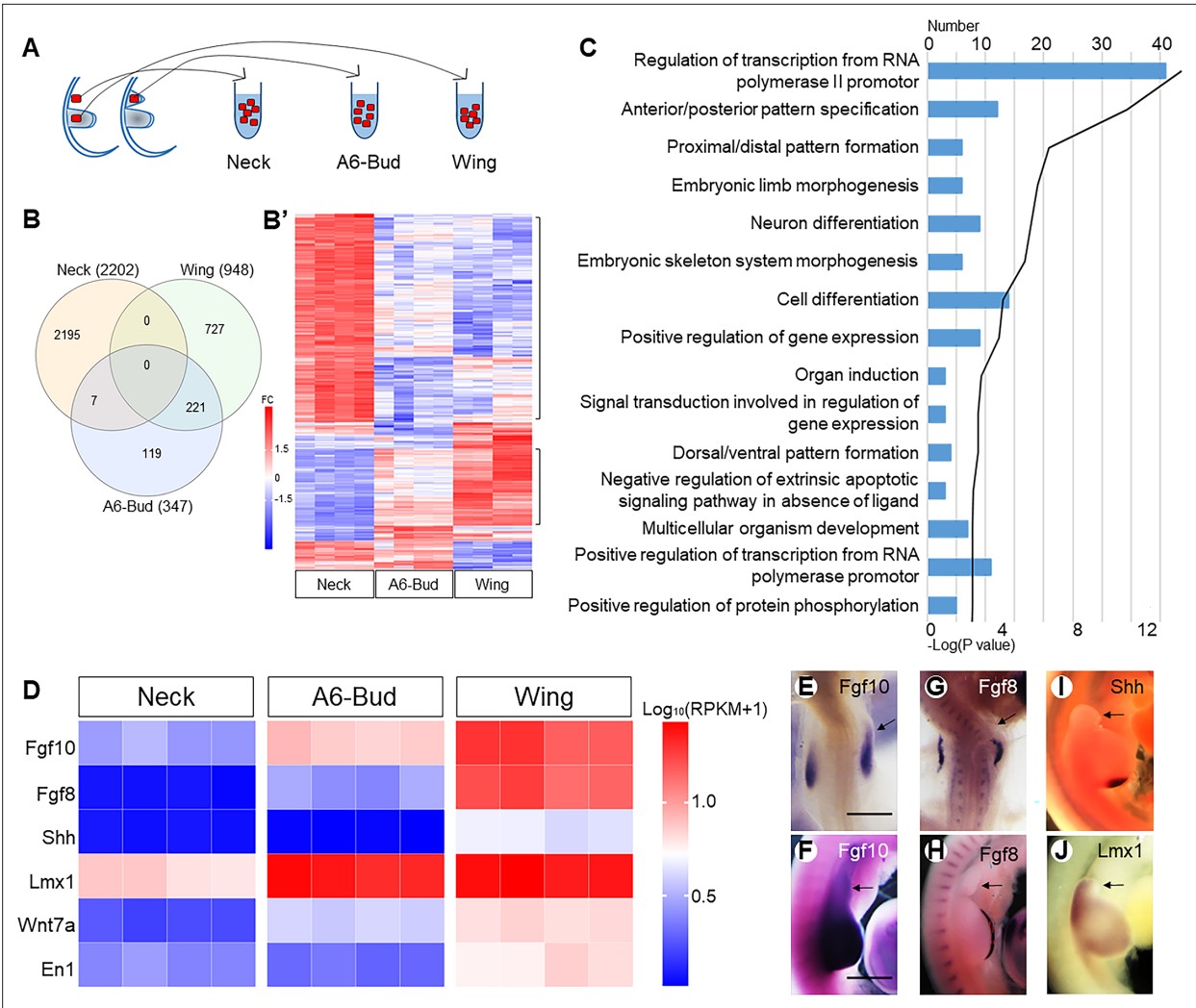

**Figure 3.** The neck wing bud is smaller than the natural wing bud. The scheme (**A**) indicates how tissue samples were collected for RNA sequencing. Venn diagram (**B**) showing the overlap between up-regulated genes expressed in normal wing bud (Wing 948), HoxA6-induced wing bud (A6-Bud 347), and neck tissue (Neck 2202) in the cervical LPM; the heatmap (**B′**) showing the expression profiles of genes in neck tissue, normal wing bud, and HoxA6-induced wing bud. FC: fold change. Gene Ontology (**C**) analyses showing top 15 terms in biological process for 221 genes of A6-Bud. The heatmap (**D**) shows the expression levels of genes related to outgrowth and patterning. The expression of *Fgf10* (**E, F**), *Fgf8* (**G, H**), *Shh* (**I**), and *Lmx1* (**J**) in transfected embryos is rechecked by ISH. (**E** and **G**): HH18-19; (**F, H, I** and **J**): HH22; scale bars in (**E**) (for **E, G**) and in (**F**) (for **F, H–J**): 500 μm. Arrows indicate induced wing buds.

between Fgf8 and Fgf10 due to the inability of Fgf signal transduction in the neck ectoderm (*Lours and Dietrich, 2005*).

## Discussion

In this study, we investigated how *Hox* genes determine the forelimb cell fate of the LPM, thus the positioning of the forelimb. We found that functional inhibition of the A-cluster *Hox4/5/6/7* genes, on the protein level, using DN forms, resulted in reduction of *Tbx5* expression and subsequently of fore-limb formation. Expression of *PG6/7* but not of *PG4/5 Hox* genes could reprogramme neck mesoderm to limb-forming mesoderm. These findings indicate different roles of *PG6/7* and *PG4/5 Hox* genes during forelimb formation.

**Table 3.** The name of 221 genes.

221 genes showed that the *A6*-induced bud closely resembles a normal wing bud.

| | Gene name |
|---|---|
| 221 Genes | ACSBG2 ALC AMPD3 ANGPTL5 AP1S2 APCDD1 APOD ASNS C4orf19 CA9 CALCA CALN1 CAMK1G CAMKK1 CASP10 CBLN3 CCDC3 CCND1 CDC7 CDH17 CG-16 CHRDL1 CKMT2 COMTD1 CRABP-I CRLF1 CRTAC1 CXCR4 CYP26C1 DACH1 DKK1 DLX5 DLX6 DNER DPYSL4 DUSP4 DUSP6 DYNC1I1 ECEL1 EDAR EGR1 EMX1 ENKUR ERMN ESM1 ETV4 ETV7 EXO1 EYA1 EYA2 FAM184B FAM222A FAM49A FGF10 FGF8 FSIP1 FSTL4 G0S2 GABRB2 GABRD GALNT17 GBX2 GJA5 GMNN GNG4 GPR176 GRIK1 GSC GSTO2 H2AFJ HES4 HGF HMP19 HOMER2 HOXA10 HOXA11 HOXA6 HOXA7 HOXA9 HOXB7 HOXC6 HOXC8 HOXC9 HOXD10 HOXD11 HOXD8 HOXD9 HPSE2 HSP90AB1 HSPE1 HTRA1 ID1 IL17RD ITPR2 JARID2 KCNAB1 KCNG1 KCNJ5 KCNT2 LDHB LGR6 LHX2 LHX9 LIMD2 LMO3 LMX1B LONRF3 LYSMD3 MAP2 MAPK11 MECOM MET MIF MSX1 MYB MYCN NEGR1 NKAIN3 NOG NPTX1 NT5E NTS OLFML1 ORC6 OVA PAX3 PCDH10 PDE3B PDGFA PFN4 PGK2 PHF24 PHLDA1 PIGA PRDM1 PRDM16 PTGS2 RAB36 RASD1 RASSF3 RASSF9 RFC3 RGS7 RSPH14 RSPO2 RTN1 RUNX3 SALL1 SCD SCG5 SCUBE1 SCUBE3 SDC1 SHOX SIM2 SLC5A1 SNAI1 SOST SOX8 SP8 SPOCK3 SPRY2 SUV39H2 TBX15 TCAIM TDO2 TEN1 TERB1 THSD7B TMEM132C TMEM132E TMEM59L TNFRSF13B TOM1L1 TOX3 TRARG1 TRMT9B TWIST3 TYW3 VEGFD WFDC1 WNT7A ZADH2 ZBTB32 ZIC2 ZIC5 ZNF385C gene:ENSGALG00000001136 gene:ENSGALG00000002461 gene:ENSGALG00000005037 gene:ENSGALG00000005790 gene:ENSGALG00000006325 gene:ENSGALG00000007131 gene:ENSGALG00000010268 gene:ENSGALG00000011040 gene:ENSGALG00000011747 gene:ENSGALG00000012045 gene:ENSGALG00000012544 gene:ENSGALG00000013268 gene:ENSGALG00000014719 gene:ENSGALG00000015366 gene:ENSGALG00000015692 gene:ENSGALG00000020895 gene:ENSGALG00000022875 gene:ENSGALG00000026154 gene:ENSGALG00000026754 gene:ENSGALG00000027002 gene:ENSGALG00000034918 gene:ENSGALG00000041500 gene:ENSGALG00000042491 gene:ENSGALG00000044224 gene:ENSGALG00000046487 gene:ENSGALG00000046504 gene:ENSGALG00000046714 gene:ENSGALG00000047687 gene:ENSGALG00000048097 gene:ENSGALG00000051549 gene:ENSGALG00000052769 gene:ENSGALG00000054625 gene:ENSGALG00000054964 gene:ENSGALG00000054968 |

## *PG4/5/6/7* genes constitute the *Hox* code activating forelimb formation

In previous genetic studies, it has been shown that, in cooperation with Wnt and retinoic acid signalling (**Nishimoto et al., 2015**), *HoxPG4/5* genes activate *Tbx5* expression (**Minguillon et al., 2012**; **Nishimoto et al., 2014**; **Moreau et al., 2019**). *Tbx5* then activates *Fgf10* expression, which leads to the thickening and epithelio-mesenchymal transition of the LPM, initiating the formation of the primary forelimb bud (**Delgado et al., 2021**; **Gros and Tabin, 2014**). Subsequently, mesodermal *Fgf10* induces ectodermal *Fgf8* expression, creating a positive feedback loop that sustains the outgrowth of the limb bud. Experiments with DN forms suggest that not only *HoxPG4/5* but also *HoxPG6/7* are required for the *Tbx5* expression in the LPM and thus for forelimb formation. Functional inhibition of any of the A-cluster of *PG4/5/6/7 Hox* genes down-regulated *Tbx5*, as well as subsequent *Fgf10* and *Fgf8* expression. The resultant lower activity of the *Fgf10-Fgf8* feedback loop ultimately limited the further development of the wing buds.

**Table 4.** Gene Ontology analyses showing top 10 terms in Biological Process.

| Wing | | A6-Bud | | Neck | |
|---|---|---|---|---|---|
| Term | p-Value | Term | p-Value | Term | p-Value |
| Regulation of transcription from RNA polymerase II promoter | 8.40E-11 | Anterior/posterior pattern specification | 2.50E-10 | Cell adhesion | 4.30E-19 |
| Anterior/posterior pattern specification | 3.20E-10 | Proximal/distal pattern formation | 8.70E-09 | Extracellular matrix organisation | 4.20E-15 |
| Embryonic skeletal system morphogenesis | 2.00E-09 | Regulation of transcription from RNA polymerase II promoter | 3.20E-08 | Transmembrane receptor protein tyrosine kinase signalling pathway | 4.80E-13 |
| Proximal/distal pattern formation | 3.70E-09 | Protein folding | 1.50E-07 | Positive regulation of kinase activity | 1.60E-10 |
| Embryonic limb morphogenesis | 8.10E-09 | Embryonic skeletal system morphogenesis | 1.30E-07 | Multicellular organism development | 3.10E-09 |
| Dorsal/ventral pattern formation | 3.40E-08 | rRNA processing | 4.90E-05 | Cell-cell adhesion | 2.30E-08 |
| Neuron differentiation | 8.30E-08 | Embryonic limb morphogenesis | 1.40E-04 | Heart development | 4.20E-08 |
| Embryonic forelimb morphogenesis | 3.70E-06 | Ribosome biogenesis | 2.70E-04 | Axon guidance | 2.00E-07 |
| Embryonic hindlimb morphogenesis | 5.50E-06 | Neuron differentiation | 4.10E-04 | Negative regulation of cell migration | 1.20E-06 |
| Multicellular organism development | 8.60E-06 | Positive regulation of gene expression | 6.40E-04 | Blood coagulation | 1.40E-06 |

In summary, our loss-of-function experiments provide direct evidence for the requirement of *PG4/5/6/7 Hox* genes for forelimb formation. Consequently, in addition to *PG4/5*, *PG6/7* genes also constitute the *Hox* code that activates the forelimb-forming programme.

## *PG6/7* genes are sufficient for forelimb formation

Ectopic expression of *HoxPG6/7* genes activated *Tbx5* expression and initiated the wing-forming programme in the neck LPM. Importantly, the induced wing bud in the neck did not grow sustainably. This may be linked to the reduced, or rather absent function of the FGF10-FGF8 feedback loop in the induced wing bud (*Cohn and Tickle, 1999*; *Yin et al., 2016*). The neck has previously been classified as a 'limb-incompetent' region, where the limb formation can only occur when both limb mesoderm and limb ectoderm are simultaneously transplanted to the neck. Transplantation of limb mesoderm alone under neck ectoderm does not support limb formation (*Lours and Dietrich, 2005*). The re-specified wing mesoderm by *HoxPG6/7* in the neck is still covered by neck ectoderm. This condition is similar to the transplantation of the prospective limb mesoderm to the neck without limb ectoderm (*Lours and Dietrich, 2005*). Since the neck ectoderm is incapable of *Fgf* signal transduction, lacking Fgf8-Fgf10 feedback loop and AER, the development of the induced wing bud stalled in the pre-AER phase.

The wing buds seen following *PG6/7* expression in the neck resemble the wing anlagen in the chicken limbless mutant, in which *Fgf8* expression is mutated and that lacks the AER and, like the induced limb buds here, the ZPA (*Grieshammer et al., 1996*; *Ros et al., 1996*; *Vogel et al., 1996*). Moreover, both the induced neck wing buds observed here and the wing buds of the limbless mutant are mainly dorsalised.

Importantly, implantation of FGF10-beads into neck LPM did not induce any wing bud structure in the neck (*Lours and Dietrich, 2005*). Neck wing buds can only be induced by ectopic expression of *HoxPG6/7* genes, as reported in the present study. This indicates that the emergence of ectopic limb buds from the neck requires re-specification of *Hox* code in the neck LPM. Despite the rudimentary outgrowth of the wing buds induced by ectopic *HoxPG6/7* expression in the neck region, our experiments demonstrate the pivotal role of *HoxPG6/7* in initiating the forelimb-forming programme.

## *PG4/5* are insufficient for forelimb formation

Although both *PG4/5* and *PG6/7 Hox* genes impinge on *Tbx5* expression, they play different roles during forelimb formation. In contrast to *HoxPG6/7*, neither the physiological expression of *HoxPG4/5* nor their overexpression in the neck region caused *Tbx5* expression and initiated formation of an ectopic wing bud. The distinct function of these two groups of *Hox* genes may be related to their expression pattern. The expression of *PG4/5* genes extends beyond the anterior border of the presumptive limb field, and some of them are expressed in the entire neck region (http://geisha. arizona.edu). Accordingly, *Tbx5* is transiently activated in the entire neck region (*Nishimoto et al., 2014*). Yet this transient activation is inadequate to initiate forelimb formation, as normally no limbs originate from the neck region. It is only in the limb field where *PG4/5* expression overlaps with expression of *PG6/7* genes that *Tbx5* expression is maintained and thus can initiate wing formation. Caudal to the forelimb region, this combinatorial effect is limited by *Hox9* expression (*Cohn et al., 1997*; *Nishimoto and Logan, 2016*; *Tanaka, 2016*). Functionally, *PG4/5 Hox* genes can activate *Tbx5* expression, but only the mesoderm expressing both *PG4/5* and *PG6/7 Hox* genes can form forelimb. Similar findings have been observed in the specification of motor neurons for the forelimb skeletal muscles (*Mukaigasa et al., 2017*). The early forelimb motor neuron programme starts in the entire neck region, but only motor neurons under the control of *Hox4/5* and *Hoxc6* complete their differentiation. Neurons solely under the control of *Hox4/5* undergo apoptosis.

## Redundancy of limb-forming *Hox* genes

The partial reduction in wing development seen after the expression of DN forms with downstream action of any of the *Hoxa4/5/6/7* genes is fully consistent with the partial redundancy among *Hox* paralog groups described for *HoxPG5* and *HoxPG6* during axial patterning (*McIntyre et al., 2007*) and for *HoxPG5* in limb development (*Xu et al., 2013*). We note, though, that we cannot formally exclude incomplete blockade of the genes targeted given the competitive nature of our approach. Be that as it may, our *Hox* inactivation experiments clearly reveal a dosage effect of *Hox* genes on

orthologous limb development. They further lead to the conclusion that normal wing development may depend on the balanced expression of *HoxPG4/5/6/7* genes.

The absence of overt limb phenotypes in PG4–PG7 mouse mutants likely reflects both the extensive functional redundancy among Hox paralogs and the difficulty of detecting subtle limb-specific effects in bilateral, systemically affected embryos. In contrast, the chick embryo system allows unilateral gene manipulation, providing an internal control and greater sensitivity for detecting weak or localised effects that may be masked in whole-animal mouse mutants. This difference in experimental sensitivity likely explains why limb phenotypes that are undetectable in mouse mutants can be clearly revealed by targeted manipulations in the chick model.

## Permissive and instructive mechanisms during limb evolution

It has been hypothesised that an interplay between permissive, instructive, and inhibitory mechanisms is needed to induce precise tissue organisation (*Morales et al., 2021*). Such an interplay may also regulate limb positioning. As shown by several authors, the caudal boundary of the forelimb is determined through the antagonism of the rostral and caudal codes: the rostral code induces forelimb formation, whereas the caudal code inhibits it (*Cohn et al., 1997*; *Moreau et al., 2019*; *Nishimoto and Logan, 2016*; *Nishimoto et al., 2014*; *Tanaka, 2016*). In the present study, we suggest that the rostral code should comprise two functionally distinct subgroups. Our data show that inhibiting HoxPG4/5 disrupts limb formation, indicating its necessity. However, overexpressing *HoxPG4/5* alone does not induce limb formation, suggesting they are not sufficient. In contrast, *HoxPG6/7* is both necessary and sufficient, as their inhibition prevents limb formation and their overexpression induces limb formation.

Thus, we speculate that *HoxPG4/5* set up a permissive environment by initiating transient *Tbx5* expression that allows limb formation to occur but does not directly trigger the process. The broad expression domain of *HoxPG4/5*, including the neck region, defines an extended permissive region where forelimb formation might be initiated. In contrast, *HoxPG6/7* instructively directs the formation of limbs by maintaining *Tbx5* expression in a precise position. The overlap of *HoxPG6/7* expression domains with the limb field further supports instructive roles of these *Hox* genes.

Moreover, the extended *Tbx5* expression domain from the heart to forelimb signifies the posterior shift of the forelimb (*Anderson et al., 2016*; *Figure 4*). As the forelimb programme proceeds, *Tbx5* expression is maintained in only the heart and the prospective forelimb region (*Figure 4*). Notably, the regression of *Tbx5* in the neck region between the heart and forelimb region implies functional differences between *PG4/5* and *PG6/7* genes.

There is an evolutionarily conserved requirement for spatial and temporal regulation of cell behaviour during morphogenesis. *Hox* codes control the growth and shape of almost all organs and the body as a whole. Therefore, the identified mechanisms by which the Hox code genes play permissive and instructive roles in controlling cell behaviour are of general significance for organogenesis during embryonic development and adult regeneration and may elucidate the regional specification mechanisms for other organs.

## Towards an evolutionary perspective of vertebral morphology and limb positioning

While the length of the cervical spinal column and the position of the forelimbs are highly fixed in mammals, they are much more variable in other vertebrates, especially from an evolutionary perspective. Indeed, there appears to be an evolutionary trend towards increased head mobility, achieved through the increasing complexity and length of the cervical spine. This trend involves, or presupposes, a caudal repositioning of the anterior limbs.

Extant jawless vertebrates such as lampreys and hagfish lack any morphological vestige, suggesting an anlage of anterior limbs. In these species, the expression of Tbx4/5, the hallmark marker of incipient anterior limb and heart development, is restricted to the latter (*Adachi et al., 2016*; *Figure 4D*). The first pectoral fins, defined by the presence of a possibly gill-arch-derived pectoral girdle (*Janvier, 1996*) and connected to the head shield, are found in fossil osteostracans, an early class of gnathostomes (*Coates, 1994*; *Figure 4A*).

The separation of the pectoral girdle from the head shield resulted in the development of a primary neck, first identifiable in placoderms (*Trinajstic et al., 2013*; *Figure 4A*). In jawed fish with paired

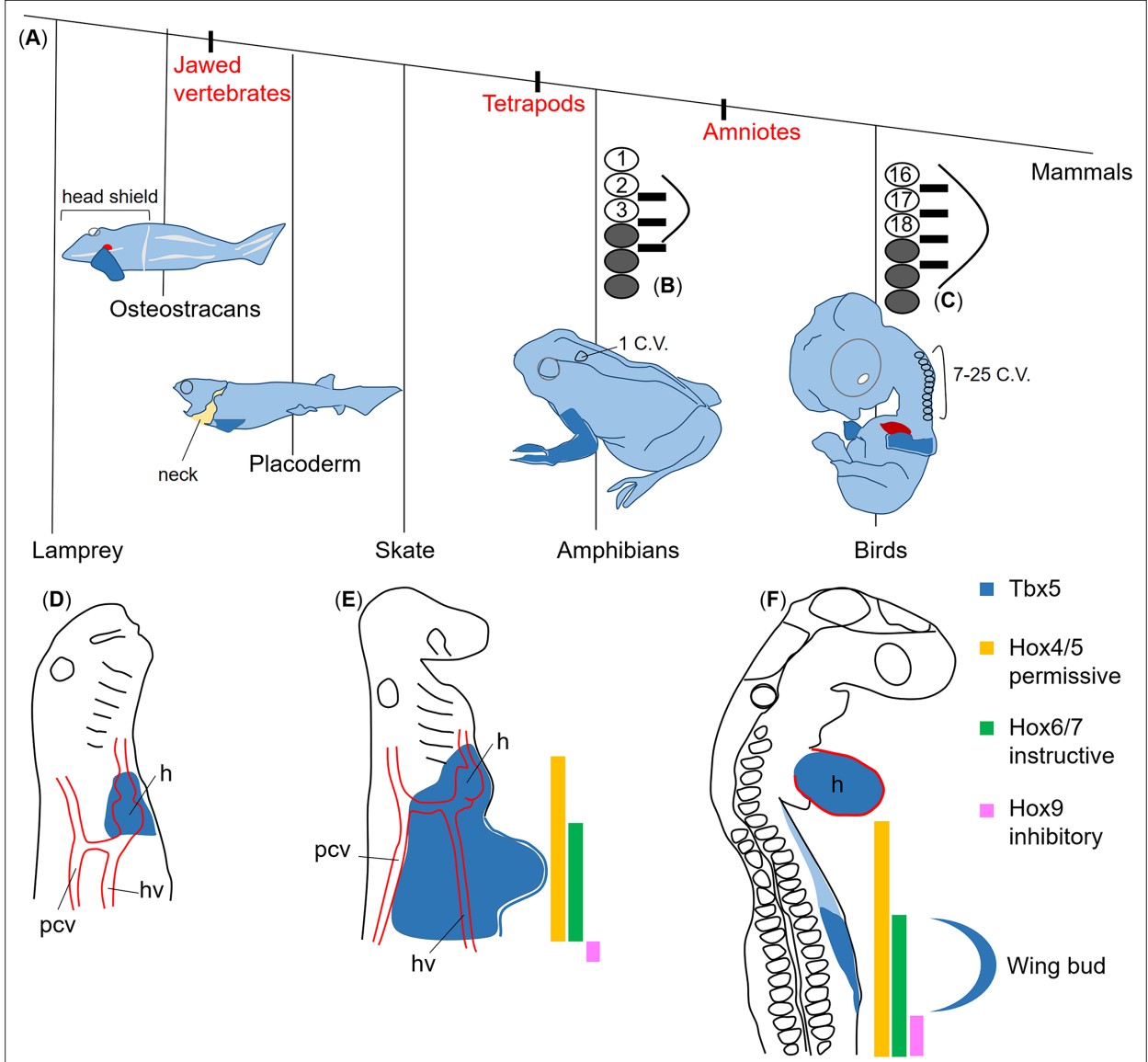

**Figure 4.** Permissive, instructive, and inhibitory *Hox* codes regulate the forelimb positioning. The phylogenetic tree of gnathostomes (**A**, redrawn from *Hirasawa et al., 2016*) shows that despite the variation in the number of cervical vertebrae (C.V.), the pectoral fin and forelimb (dark blue) are always located at the cervical-thoracic boundary. However, their axial positions with respect to somite number vary widely across species (**B** and **C** modified from *Burke et al., 1995*). (**B**, **C**) Bright circles: numbered somites; grey shaded circles: thoracic somites; black bars: spinal nerves of the brachial plexus; curved lines: limb bud. In lamprey embryos (**D**), expression of *Tbx5* homologue is restricted to the heart region (*Adachi et al., 2016*). In skate embryos (**E**), *Tbx5* expression (blue) extended slightly caudally from the heart anlage (*Adachi et al., 2016*). In avian embryos (**F**), it extends from the heart over the neck to the wing field. During wing bud formation, *Tbx5* expression is restricted to only the heart and the wing bud. In lateral plate mesoderm, *Hox4/5* expression (yellow) extends into the neck region, whereas the anterior expression domain of *Hox6/7* (green) is at the wing level. *Hox9* expression (magenta) starts posteriorly to the wing. The yellow, green, and magenta colours represent permissive, instructive, and inhibitory functions, respectively. The blue curved line outlines the wing bud.

fins, the evolutionary caudal repositioning of the anterior pectoral fins can also be verified by the fact that the expression of Tbx5 is now slightly caudal to the heart anlage (*Anderson et al., 2016*). This has been documented in skates, as well as zebrafish (*Adachi et al., 2016*; *Criswell et al., 2021*; *Figure 4E*).

A true neck connecting the cranium and trunk first evolved in amphibians—the first land vertebrates—as the pectoral girdle shifted caudally and the first trunk vertebra transformed into a cervical vertebra (*Torrey, 1978*; *Figure 4A and B*). With the further evolution of land vertebrates, the number of cervical vertebrae increased significantly (*Goodrich, 1906*). The longest cervical vertebral columns,

with 76 segments, have been reported in the fossil diapsids Muraenosaurus and Elasmosaur Albertonectes (*Kubo et al., 2012*). In birds, the number of cervical vertebrae varies widely, ranging from nine to twenty-five (*Yapp and Lyons, 1965*; *Figure 4A, C, and E*). The evolutionarily retained muscular connection between the head and shoulder girdle, formed by the cucullaris muscle and its derivatives, validates this history (*Sefton et al., 2016*; *Theis et al., 2010*).

The significance of Hox genes in vertebrate diversification and limb complexity has been repeatedly documented (*Cohn and Tickle, 1999*; *Wellik and Capecchi, 2003*; *Li et al., 2023*; *Kort and Polly, 2023*). The present results refine our understanding of how *Hox* genes integrate vertebral column structure and limb positioning, which together have led to the extensive behaviour and foraging/predatory diversification of vertebrates (*Rytel et al., 2024*; *Marek et al., 2021*).

## Materials and methods

### In ovo electroporation

Fertilised chicken (*Gallus gallus domesticus*) eggs were obtained from the Institute of Animal Sciences of the Agricultural Faculty, University of Bonn, Germany. First, after windowing of the egg shell and exposing the embryo, a solution containing 5–10 µg/µL plasmid and 0.1% Fast Green was injected into the coelom at specific axial levels. Electroporation was then performed using the CUY 21-Edit-II electroporator with one poration pulse of high voltage (0.01 ms, 70 V) followed by two driving pulses of low voltage (50 ms, 7 V, with 200 ms intervals). There is a 99.9 ms interval between the high- and low-voltage pulses. After re-incubation, embryos were imaged under the Nikon SM21500 fluorescence microscope and then fixed in 4% paraformaldehyde overnight at 4°C.

### Plasmids for electroporation

DNA plasmids were produced by Dongze Bio-products (Guangzhou, China). Coding sequences (obtained from NCBI) for Hoxa4 (930 bp, NM_001030346.3), Hoxa5 (813 bp, NM_001318419.2), Hoxa6 (696 bp, NM_001030987.4), Hoxb6 (669 bp, NM_001396636.1), Hoxc6 (714 bp, NM_001407494.1), Hoxa7 (660 bp, NM_204595.3), or Hoxb7 (654 bp, XM_040653307.2) were inserted into the pCAGGS-P2A-EGFP plasmid. A plasmid expressing the dominant-negative (dn) form specific for Hoxa4, a5, a6, or a7 was produced using their coding sequence lacking the C-terminal portion, including Hoxa4dn (762 bp), Hoxa5dn (729 bp), Hoxa6dn (585 bp), and Hoxa7dn (528 bp). A large quantity of DNA plasmids was purified using the NucleoBond Xtra Midi DNA preparation kit (Macherey-Nagel).

### RNA ISH

Whole-mount RNA ISH was performed by incubating probes at 65°C. The probes were detected using anti-Digoxigenin-AP, fab fragments (Roche), and colour reagent NBT/BCIP staining solution (Roche). Chicken Lmx-1, Fgf10, and Fgf8 probes were provided by H Ohuchi, O Pourquie, and C Tabin, respectively. Chicken Tbx5 probe, Hox probes, and Hoxdn C-terminal probes were produced using PCR and transcribed using the DIG-RNA Labelling Kit (Roche, #11175025910) with T7 polymerase. The specific primers were shown in *Table 5*.

### RNA sequencing analyses

Wing parts (five samples per replicate, four replicates, total 20 samples) and neck parts (20 samples per replicate, four replicates, total 80 samples) were dissected from HH22 normal embryos. Additionally, a total of 80 *Hoxa6*-induced ectopic buds (20 samples per replicate, four replicates) were dissected from

**Table 5.** Primer sequences used for generating in situ hybridisation probes by PCR.

| Genes | Forward primer | Reverse primer |
|---|---|---|
| Tbx5 | TACTGGAGCCCACTGGATGA | ATGCTCGGTGGTGGAACATT |
| Hoxa4 | ATGACCATGAGTTCGTTTTTGAT | GCTAGCGCGGCCGCGT |
| Hoxa5 | TGAAAAACTCCCTGGGCAACTC | AGCTGCCATGCTCATACTTTTC |
| Hoxa6 | CAGTCCAACACCGTCATTGC | CTCCCCTGACTTTTCCTCTGTT |
| Hoxa7 | TCAAAGCCCGTTCTCTTCCG | AGATCTTGATCTGCCGCTCC |

HH22 embryos with *Hoxa6* ectopic expression in the neck. The dissections were performed under the Nikon SM21500 fluorescence microscope. Only ectopic buds identified by their morphology and EGFP expression were isolated and collected, including the surface ectoderm. Total RNA of samples was isolated with the miRNeasy Micro Kit (QIAGEN). Library preparation was performed according to the manufacturer's protocol using the 'VAHT Universal RNA-Seq Library Prep Kit for Illumina V6 with mRNA capture module'. Next, 500 ng total RNA was used for mRNA capturing, fragmentation, cDNA synthesis, adapter ligation, and library amplification. Bead-purified libraries were normalised and finally sequenced on the HiSeq 3000/4000 system (Illumina Inc, San Diego, CA, USA).

## Statistical analysis

Data analyses on FASTQ files were conducted with CLC Genomics Workbench (version 21.0.4, QIAGEN, Venlo, NL). The reads of all probes were adapter-trimmed (Illumina TruSeq) and quality-trimmed. Mapping was done against the *Gallus gallus* (GRCg6a) (19 March 2021) genome sequence. Statistically significant differential expression was determined using the 'Differential Expression for RNA-Seq' tool (version 2.4) (QIAGEN Inc 2021). The resulting p-values were corrected for multiple testing by FDR. The RNA expression level was indicated by reads per kilobase of transcript per million mapped reads (RPKM), and the statistical analysis between the three groups was made by ordinary one-way ANOVA, using GraphPad Prism v6 (San Diego, CA, USA). Functional annotation clustering was done by means of the DAVID online tool (https://david.ncifcrf.gov/) and using the GO 'biological process' annotation category. Data are presented as mean ± standard error of the mean. The level of statistical significance was set at **$p<0.01$.

## Acknowledgements

The authors thank Dr Frank Stockdale for helpful discussions and valuable comments on the manuscript. We thank Sandra Graefe and Heinz Bioernsen for their expert technical assistance. Computational support from the Centre for Information and Media Technology, especially the High-Performance Computing team at Heinrich-Heine University, is acknowledged. This work was supported by grants from China Scholarship Council (CSC) and by German Research Funding (DFG-Hu 729/13).

## Additional information

### Funding

| Funder | Grant reference number | Author |
| --- | --- | --- |
| Deutsche Forschungsgemeinschaft | DFG-Hu 729/13 | Ruijin Huang |
| China Scholarship Council | | Yajun Wang |

The funders had no role in study design, data collection and interpretation, or the decision to submit the work for publication.

### Author contributions

Yajun Wang, Conceptualization, Data curation, Formal analysis, Funding acquisition, Validation, Investigation, Writing – original draft; Maik Hintze, Ketan Patel, Qin Pu, Conceptualization, Writing – review and editing; Jinbao Wang, Hengxun Tao, Patrick Petzsch, Karl Köhrer, Longfei Cheng, Peng Zhou, Jianlin Wang, Zhaofu Liao, Xu-Feng Qi, Dongqing Cai, Methodology; Thomas Bartolomaeus, Conceptualization; Karl Schilling, Joerg Wilting, Georgy Koentges, Writing – review and editing; Stefanie Kuerten, Ruijin Huang, Conceptualization, Funding acquisition, Writing – review and editing

### Author ORCIDs

Yajun Wang (iD) https://orcid.org/0009-0004-6955-0454
Xu-Feng Qi (iD) https://orcid.org/0000-0002-5911-071X
Ketan Patel (iD) https://orcid.org/0000-0002-7131-749X
Ruijin Huang (iD) https://orcid.org/0000-0003-0467-2907

## Ethics

Ethical approval was not received for this animal study because According to German Ordinance on the Protection of Animals Used for Scientific Purposes (Tierschutzversuchstierverordnung), permission for in ovo experiments on avian embryos is not required. Although no permission for studies on in ovo development is reuqired, we carry out the experiment very carfully and according to the 3R prinziples.

Reviewer #2 (Public review): https://doi.org/10.7554/eLife.100592.3.sa1
Author response https://doi.org/10.7554/eLife.100592.3.sa2

---

# Additional files

## Supplementary files

MDAR checklist

## Data availability

Source data has been uploaded to Dryad (https://doi.org/10.5061/dryad.cz8w9gjk5).

The following dataset was generated:

| Author(s) | Year | Dataset title | Dataset URL | Database and Identifier |
|---|---|---|---|---|
| Wang Y, Huang R | 2026 | Permissive and instructive Hox codes govern limb positioning | https://doi.org/10.5061/dryad.cz8w9gjk5 | Dryad Digital Repository, 10.5061/dryad.cz8w9gjk5 |

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
