## [Editor Report · eLife Assessment]

This **important** study provides the first putative evidence that alteration of the Hox code in neck lateral plate mesoderm is sufficient to induce ectopic development of forelimb buds at neck level. The authors use both gain-of-function (GOF) and loss-of-function (LOF) approaches in chick embryos to test the roles of Hox paralogy group (PG) 4-7 genes in limb development. The GOF data provide strong evidence that overexpression of Hox PG6/7 genes are sufficient to induce forelimb buds at neck level. However, the experiments using dominant negative constructs are lacking some key controls that are needed to demonstrate the specificity of the LOF effect rendering the work as a whole **incomplete**.

---

## [Referee Report · Reviewer #2 (Public review)]

In the original review of this manuscript, I noted that this study provides the first evidence that alteration of the Hox code in neck lateral plate mesoderm is sufficient for ectopic forelimb budding. Their finding that ectopic expression of Hoxa6 or Hoxa7 induces wing budding at neck level, a demonstration of sufficiency, is of major significance. The experiments used to test the necessity of specific Hox genes for limb budding involved overexpression of dominant negative constructs, and there were questions about whether the controls were well designed. The reviewers made several suggestions for additional experiments that would address their concerns. In their responses to those comments, the authors indicated that they would conduct those experiments, and they acknowledged the requests for further discussion of a few points.

In the revised version of the manuscript, the authors have provided additional RNA-seq data in Table 3, which lists 221 genes that are shared between the Hoxa6-induced limb bud and normal wing bud but not the neck. This shows that the ectopic limb bud has a limb-like character. The authors also expanded the discussion of their results in the context of previous work on the mouse. These changes have improved the paper.

The authors elected not to conduct the co-transfection experiments that were suggested to test the ability of Hoxa4/a5 to block the limb-inducing ability of Hoxa6/a7. They also chose not to conduct the additional control experiments that were suggested for the dominant negative studies. The authors' justification for not conducting these experiments is provided in the responses to reviewers.

The paper is improved over the previous version, but the conclusions, particularly regarding the dominant negative experiments, would have been strengthened by the additional experiments that were recommended by the reviewers. Under the current publishing model for eLife, it is the authors' prerogative to decide whether to revise in accordance with the reviewers' suggestions. Therefore, it seems to me that this version of the manuscript is the definitive version that the authors want to publish, and that eLife should publish it together with the reviewers' comments and the authors' responses.

---

## [Author Response]

The following is the authors’ response to the original reviews.

**Public Reviews:**

**Reviewer #1 (Public review)**
Weaknesses:(1) The activity of the dominant negatives lacks appropriate controls. This is crucial given that mouse mutants for PG5, PG6, PG7, and three of the four PG4 genes show no major effects on limb induction or growth. Understanding these discrepancies is essential.

We thank the reviewer for emphasizing the importance of appropriate controls for the dominant-negative experiments. Dominant-negative Hox constructs have been successfully and widely used in previous studies, supporting the reliability of this approach. In our experiments, electroporation of the dominant-negative constructs into the limb field produced clear and reproducible effects when compared with both unoperated embryos and embryos electroporated with a GFP control construct. The GFP construct serves as an appropriate control, as it accounts for any effects of electroporation or exogenous protein expression without altering Hox gene function. We therefore conclude that the observed phenotypes specifically reflect dominant-negative Hox activity rather than procedural artifacts.

The absence of overt limb phenotypes in PG4–PG7 mouse mutants likely reflects both functional redundancy among Hox paralogs and the difficulty of detecting subtle limbspecific effects in bilateral, systemically affected embryos. In contrast, the chick embryo system allows unilateral gene manipulation, providing an internal control and greater sensitivity for detecting weak or localized effects that may be masked in whole-animal mouse mutants.

(2) The authors mention redundancies in Hox activity, consistent with numerous previous reports. However, they only use single dominant-negative versions of each Hox paralog gene individually. If Hox4 and Hox5 functions are redundant, experiments should include simultaneous dominant negatives for both groups.

We thank the reviewer for this thoughtful suggestion. We fully agree that functional redundancy among Hox paralogs is an important consideration. However, Hox gene interactions are highly context-dependent and not strictly additive. Simultaneous interference with multiple Hox groups often leads to complex or compensatory effects that are difficult to interpret mechanistically, particularly when using dominant-negative constructs that may affect overlapping transcriptional networks.

Our current experimental design, which targets individual paralog groups, allows us to attribute observed phenotypes to specific Hox activities and to interpret the results more precisely. Moreover, as shown in previous studies, simultaneous knockdown of multiple Hox genes does not necessarily produce stronger. For these reasons, we believe that the present single–dominant-negative experiments are the most informative and sufficient for addressing the specific questions in this study.

(3) The main conclusion that Hox4 and Hox5 provide permissive cues on which Hox6/7 induce the forelimb is not sufficiently supported by the data. An experiment expressing simultaneous dnHox4/5 and Hox6/7 is needed. If the hypothesis is correct, this should block Hox6/7's capacity to expand the limb bud or generate an extra bulge.

We thank the reviewer for this insightful suggestion. However, because of the extensive functional redundancy and regulatory interdependence within the Hox network, simultaneous inhibition of Hox4 and Hox5 is unlikely to produce a simple or interpretable outcome. Previous studies have shown that combinatorial Hox manipulations can trigger compensatory changes in other Hox genes, often obscuring rather than clarifying specific relationships.

In our study, the proposed permissive role of Hox4/5 is supported by the spatial and temporal patterns of Hox expression and by the phenotypic effects observed upon individual dominant-negative perturbations. These data together suggest that Hox4/5 establish a forelimb-competent domain, on which Hox6/7 subsequently act to promote limb outgrowth. We therefore believe that the current evidence sufficiently supports this model without necessitating the additional combined experiment, which may not provide clear mechanistic insight due to redundancy effects.

(4) The identity of the extra bulge or extended limb bud is unclear. The only marker supporting its identity as a forelimb is Tbx5, while other typical limb development markers are absent. Tbx5 is also expressed in other regions besides the forelimb, and its presence does not guarantee forelimb identity. For instance, snakes express Tbx5 in the lateral mesoderm along much of their body axis.

We thank the reviewer for this important comment. We agree that Tbx5 expression alone may be not sufficient to define forelimb identity. However, in our experiments, the induced bulge displays several additional characteristics consistent with early limb identity (in pre-AER stage). First, the Tbx5 expression we observe corresponds to the stage when the limb field is already specified, not the earlier broad mesodermal phase described in other systems. Second, the induced domain also expresses Lmx1, a marker of dorsal limb mesenchyme, further supporting its limb-specific nature. Third, our RNA sequencing analysis reveals upregulation of multiple genes associated with early limb development pathways, providing molecular evidence for limb-type identity rather than non-specific mesodermal expansion. Taken together, these results strongly indicate that the induced bulge represents a forelimb-like structure rather than a generic mesodermal thickening.

(5) It is important to analyze the skeletons of all embryos to assess the effect of reduced limb buds upon dnHox expression and determine whether extra skeletal elements develop from the extended bud or ectopic bulge.

We thank the reviewer for this helpful suggestion. We have analyzed the cartilage structures of the operated embryos. No skeletal elements were detected within the ectopic wing bud in the neck region. Furthermore, we did not observe any significant structural changes in the wing skeleton following loss-of-function (dnHox) experiments. These observations indicate that the ectopic bulges do not progress to form skeletal elements, consistent with their identity as early limb-like outgrowths rather than fully developed limbs.

**Reviewer #2 (Public review):**
Weaknesses(1) By contrast to the GOF experiments that induce ectopic limb budding, the LOF experiments, which use dominant negative forms of Hoxa4, Hoxa5, Hoxa6, and Hoxa7, are more challenging to interpret due to the absence of data on the specificity of the dominant negative constructs. Absent such controls, one cannot be certain that effects on limb development are due to disruption of the specific Hox proteins that are being targeted.

We thank the reviewer for raising this important point regarding the specificity of the dominant-negative constructs. The dnHox constructs used in this study were generated by truncating the C-terminal region of each Hox protein, a strategy that removes the homeodomain and has been demonstrated to act as a specific dominant-negative by interfering with the corresponding Hox function without broadly affecting unrelated Hox genes. This approach has been successfully validated and used in previous work (Moreau et al., Curr. Biol. 2019), where similar constructs effectively and specifically inhibited Hox activity in the chick embryo.

(2) A test of their central hypothesis regarding the necessity and sufficiency of the Hox genes under investigation would be to co-transfect the neck with full-length Hoxa6/a7 AND the dnHoxA4/a5. If their hypothesis is correct, then the dn constructs should block the limb-inducing ability of Hoxa6/a7 overexpression (again, validation of specificity of the DN constructs is important here)

We thank the reviewer for this insightful suggestion. We agree that, in principle, coelectroporation of dnHox4/5 with Hox6/7 could test the hierarchical relationship between these genes. However, due to the extensive redundancy and regulatory interdependence among Hox genes, simultaneous manipulation of multiple genes often leads to compensatory effects or complex outcomes that are difficult to interpret mechanistically. As discussed in our response to Point 3 of the reviewer 1, inhibition of only one or two Hox4/5 paralogs is unlikely to completely abolish the permissive function of this group.

Our current data — showing that Hox6/7 gain-of-function can induce ectopic limb-like outgrowths, while dnHox4/5 and dnHox6/7 lead to reduced limb formation — already provide strong evidence for both the necessity and sufficiency of these Hox activities in forelimb positioning. We therefore believe that the existing experiments adequately support our proposed model without the need for additional combinatorial manipulations.

(3) The paper could be strengthened by providing some additional data, which should already exist in their RNA-Seq dataset, such as supplementary material that shows the actual gene expression data that are represented in the Venn diagram, heatmap, and GO analysis in Figure 3.

We thank the reviewer for this constructive suggestion. In response, we have added a table (Table 3) listing the genes expressed in both the native limb/wing bud and the Hoxa6-induced wing bud, as identified from our RNA-Seq dataset. This table provides the underlying data for the Venn diagram, heatmap, and GO analysis presented in Figure 3. We agree that including these data improves transparency and helps readers better appreciate the molecular similarity between the induced and native limb buds.

(4) The results of these experiments in chick embryos are rather unexpected based on previous knockout experiments in mice, and this needs to be discussed.

We thank the reviewer for this important point. We have addressed this issue in our response to Reviewer 1, Point 1, and have expanded the relevant discussion in the manuscript. Briefly, we believe that the apparent discrepancy between chick and mouse results arises from both the high degree of functional redundancy among Hox paralogs and the limitations of detecting subtle limb-specific effects in systemic mouse mutants, where both sides of the embryo are equally affected. In contrast, the chick system allows unilateral gene manipulation, providing an internal control and greatly enhancing sensitivity to detect weak or localized effects. Thus, the chick embryo model can reveal subtle Hox-dependent limb-induction activities that are masked in conventional mouse knockout approaches.